# Molecular Multi-Target Approach for Human Acetylcholinesterase, Butyrylcholinesterase and *β*-Secretase 1: Next Generation for Alzheimer’s Disease Treatment

**DOI:** 10.3390/ph16060880

**Published:** 2023-06-15

**Authors:** Géssica Oliveira Mendes, Samuel Silva da Rocha Pita, Paulo Batista de Carvalho, Michel Pires da Silva, Alex Gutterres Taranto, Franco Henrique Andrade Leite

**Affiliations:** 1Laboratory of Molecular Modeling, Department of Health, State University of Feira de Santana, Salvador 44036-900, BA, Brazil; gomendes05@gmail.com; 2Postgraduate Program in Pharmaceutical Sciences, State University of Feira de Santana, Salvador 44036-900, BA, Brazil; samuel.pita@ufba.br; 3Laboratory of Bioinformatics and Molecular Modeling (LaBiMM), Pharmacy College, Federal University of Bahia (UFBA), Salvador 40170-110, BA, Brazil; 4Feik School of Pharmacy, University of the Incarnate Word, San Antonio, TX 78212, USA; pcarvalh@uiwtx.edu; 5Laboratory of Bioinformatics and Drug Design, Department of Bioengineering, Federal University of Sao Joao del-Rei, São João del-Rei 36301-1601, MG, Brazil; michel@cefetmg.br (M.P.d.S.); proftaranto@hotmail.com (A.G.T.); 6Federal Center for Technological Education of Minas Gerais, Department of Informatics, Management and Design, R. Álvares de Azevedo, 400, Bela Vista, Divinópolis 35503-822, MG, Brazil

**Keywords:** molecular hybrids, Alzheimer’s disease, cholinesterase, human *β*-secretase 1, molecular dynamics

## Abstract

Alzheimer’s Disease (AD) is a neurodegenerative condition characterized by progressive memory loss and other affected cognitive functions. Pharmacological therapy of AD relies on inhibitors of the enzymes acetylcholinesterase (AChE) and butyrylcholinesterase (BChE), offering only a palliative effect and being incapable of stopping or reversing the neurodegenerative process. However, recent studies have shown that inhibiting the enzyme β-secretase 1 (BACE-1) may be able to stop neurodegeneration, making it a promising target. Considering these three enzymatic targets, it becomes feasible to apply computational techniques to guide the identification and planning of molecules capable of binding to all of them. After virtually screening 2119 molecules from a library, 13 hybrids were built and further screened by triple pharmacophoric model, molecular docking, and molecular dynamics (t = 200 ns). The selected hybrid G meets all stereo-electronic requirements to bind to AChE, BChE, and BACE-1 and offers a promising structure for future synthesis, enzymatic testing, and validation.

## 1. Introduction

Alzheimer’s disease (AD) is a progressive, irreversible, and insidious neurodegenerative disease that causes memory loss and various cognitive disorders [1]. AD is responsible for about 70% of all cases of dementia worldwide, with a current prevalence of 24 million people and estimated to double by 2040 [2,3].

The pathogenesis seems to involve three main factors. (i) Deficiency in cholinergic transmission caused by the loss of cholinergic neurons; (ii) accumulation of extracellular deposits of β-amyloid protein due to the catalytic action of β-secretase 1; (iii) formation of neurofibrillary clusters of a phosphorylated form of the TAU protein [3,4,5].

The cholinergic hypothesis is described as one of the causes of AD [3]. Cholinesterases are a family of enzymes responsible for catalyzing the hydrolysis of acetylcholine (ACh) into choline and acetic acid, and are divided into AChE and BChE [6,7]. These enzymes are mainly found in the central nervous system (CNS) and their inhibition would lead to an increase in ACh available in nerve endings, promoting cognitive improvement [7]. In AD patients, AChE activity may remain unchanged or decline. When it declines, BChE activity progressively increases, thus generating an imbalance between them [8,9].

The amyloid hypothesis relates AD to the formation of extracellular deposits of β-amyloid peptide and the accumulation of insoluble plaques in neurons. This pathophysiological process involves the enzyme β-secretase (BACE-1) initiating the cleavage of the transmembrane protein (Amyloid Precursor Protein-APP). This cleavage is terminated by another enzyme, γ-secretase, generating the β-amyloid peptide (Aβ), which aggregates into oligomers, forming plaques that are deposited in different parts of the brain, mainly in neurons of the hippocampus, basal nucleus, cortex entorhinal and associative cortex [10]. One of the few drugs currently being used to address the amyloid hypothesis is the recently FDA-approved Aduhelm© (Aducanumab) [11], a drug that reduces extracellular deposits of β-amyloid peptide. It is a monoclonal antibody that targets aggregated forms of amyloid beta agglomerates and reduces its buildup.

Despite its relevance in epidemiological and public health terms, the drug repertoire against AD is limited and associated with serious adverse effects such as hepatotoxicity, hypertension, and weight loss [3]. These limitations make it urgent to search for new drugs capable of preventing disease progress, controlling symptoms, and increasing patients’ survival and quality of life [3,12].

Computer-aided drug design (CADD), one of the modern tools used in the development of new drugs, is used to investigate the chemical interactions of a ligand (drug) with its receptor in the exploration of structural factors related to the biological effect and prediction of potential pharmacological activities [13]. Among the available CADD techniques, Virtual Receptor-Based Screening uses knowledge of the three-dimensional (3D) structure of disease-related targets (receptors, enzymes) to select molecular structures with a good level of complementarity with these targets and respective binding affinity or molecular docking intensity [14].

Recent studies focus on the identification of multi-targeted drugs for the treatment of AD patients with approaches based on classical strategies, such as synthesis and biological evaluation of molecular hybrids [15,16]. Additionally, the application of computational techniques that take into account the total flexibility between the ligand and the macromolecule, such as molecular dynamics studies, allow us to describe the variation in molecular behavior as a function of time, with a greater approach to biological conditions [17,18].

Another strategy that can be also employed is the pharmacophore-based approach, where multi-target molecules are built from the union of pharmacophores, that is, by molecular hybridization, in which a new chemical entity is formed by the union of two or more pharmacophoric units of bioactive molecules through linkers, using groups that are not cleaved, and thus act on the targets. Multitarget molecules are built from the preservation of important groups [15,19,20].

In this work, we present the identification of molecular fragments with high-binding efficiency (ΔG < −0.4 Kcal/mol) to AChE, BChE, and BACE-1 and subsequent docking through molecular hybridization, with the proposed hybrids evaluated through virtual screening (molecular docking and pharmacophore model) and molecular dynamics simulations.

## 2. Methods

### 2.1. Docking-Based Virtual Screening

The fragment bank available for purchase (Fragmenta) and present on the ZINC15 platform [21] was hierarchically subjected to molecular docking against the 3D structures of the targets, AChE, BChE and BACE-1, respectively.

The 3D structures of the targets, AChE, BChE, and BACE-1, were obtained from the Protein Data Bank (PDB) under code 4M0E, 4BDS and 6UWP, respectively, using resolution value and the presence of ligand as search parameters. Target structures were prepared with the biopolymer module implemented in SYBYL-X 2.0 [22], where ions and water were removed and hydrogen atoms were inserted to optimize hydrogen bonds. For the AChE and BChE target structures, the protonation state of the receptors was adjusted to pH 7.4 through PropKa [23] server, and the conformational search and scoring were performed by AutoDock Vina 1.1.2 [24], according to previously validated parameters [25]. As for the BACE-1 structure, the receptor had its protonation state evaluated by the H++ 1.0 server program and pKa corrected at pH 4.5 [26]. Validation methods were used, and after obtaining the data, the program selected for molecular docking with BACE-1 was GOLD 5.8.1 [27,28]. The score was provided by the Astex Scoring Potential (ASP, knowledge-based function derived from a database of protein–ligand complexes) function with the parameters previously validated [29]. The search parameters were evaluated for the ability to identify the crystallographic pose through redocking with the better pose according to root-mean-square deviation (RMSD < 2 Å) and asset recognition capability against false positives by calculating the area under the ROC curve (AUC > 0.7) [25,29].

Initially, molecules were classified based on the affinity energy provided by the programs. Those with energies greater than the median were analyzed for binding efficiency (LE < −0.4 Kcal/mol) with the aid of Marvin© Sketch 15.4.20 [30] to calculate the number of heavy atoms, according to Equation (Equation 1) [25].
(1)$$BE=\frac{\Delta G}{N}$$
where:

BE = Binding Efficiency

ΔG = Affinity Energy

*N* = Number of heavy atoms 

Binding efficiency provides an estimation of physicochemical properties that can be optimized to increase affinity for a target. The binding efficiency metric is recommended at all stages of drug discovery, starting with fragment selection or a screening success [31]. Then, the binding efficiency is the ratio between the affinity energy value and the number of heavy atoms, with the fragments that have a BE less than −0.4 Kcal/mol being selected. Following this, the fragments selected were submitted to an analysis of interactions with the main amino acids present in the orthosteric site of the crystallographic structure of the targets.

### 2.2. Molecular Hybridization

The most active thirteen fragments (the fragment bank available for purchase (Fragmenta) and present on the ZINC15 platform) against the selected targets were joined by adding linkers [15,32], using the program Marvin© Sketch 15.4.20 [30]. It is worth noting that each union of the fragments was selected with respect to affinity towards the targets. Thus, the best fragments for each target were selected and united, forming hybrid molecules designed to bind to the three targets. These hybrids were converted to 3D format and minimized on the SYBYL-X${}^{©}$ platform. 2.0 [22].

### 2.3. Physicochemical Filters

The hybrids designed were characterized using the pkCSM server [33] for the physicochemical descriptors in Lipinski’s rule and evaluated potential mutagen (*Salmonella typhimurium* reverse mutation assay-AMES test), namely: the number of hydrogen bond acceptors (≤10), hydrogen bond donor groups (≤5), molecular mass (≤500 g/mol), octanol–water partition coefficient (clogP) (≤5) and polar surface area (PSA) (≤140 Å${}^{2}$) and Veber’s rules [34].

The AMES test was performed against *Salmonella typhimurium* and *Escherichia coli*. It is based on the knowledge that if a substance is mutagenic for these bacteria, it also presents a carcinogenic risk in humans [35]. Physicochemical characteristics suggesting oral bioavailability are also evaluated, as these drugs are intended mostly for elderly patients with chronic diseases. Such patients benefit from orally available drugs, as they tend to increase patient adherence to the treatment.

### 2.4. Pharmacophore-Based Virtual Screening

A pharmacophoric model of triple inhibitors previously developed by our group [16] was used to filter the molecular hybrids. This step was implemented through the flexible 3D alignment routine, available in the UNITY 3D module on the SYBYL-X© platform. 2.0 [22]. The quality of the alignment of the molecules will be expressed by their QFIT value, which can vary from 0 to 100. The QFIT value may be regarded as a measure of the degree of closeness with which the hit pharmacophore matches the corresponding query feature coordinates within a given range of a spatial constraint tolerance. Where, the closer the fit, the higher the value [36].

### 2.5. Molecular Docking

The hybrids selected in the previous step were subjected to molecular docking, AutoDock Vina 1.1.2 [24] was used for the AChE and BChE [24] targets; whereas GOLD 5.4.0 [28] was used for BACE-1 [25] (following the same protocol and steps as 2.1) to obtain the pose with best affinity, analyze the interaction of the hybrids at the binding site under test and also to generate the topology that was used in the molecular dynamics.

### 2.6. Molecular Dynamics

#### 2.6.1 Parameterization of the Ligand

The hybrid topology prioritized through molecular docking assays was generated according to the ATB 3.0 server [37]. Atomic charge, bond length, torsional angles, and dihedral parameters were obtained using the GROMOS96 54A7 force field [38].

#### 2.6.2 MD Simulations 

MD simulations were performed in the GROMACS 5.1.2 package [39]. The APO forms were previously obtained [33]. In this step, 3D structures were obtained from the PDB (AChE: 4M0E; BChE: 4BDS; BACE-1: 6UWP), which the crystallographic ligand, non-structural water, and crystallization artifacts were removed. The unmodeled regions of the structures were built using the SWISS-MODEL server [40]. The protonation state of acidic and basic residues of the targets was adjusted in the pdb2gmx module implemented in GROMACS 5.1.2 [39] according to pH 7.4 for AChE and BChE [25] and pH 4.5 for BACE-1 [26]. The pKa values of the residuals were evaluated by the H++ server, except the catalytic residues of BACE-1, which were manually adjusted for the protonated (Asp32) and deprotonated (Asp228) state [41]. A dodecahedral box with the SPC-E water model was used [42] to solvate the system, with a minimum distance of 1.4 nm from the edge of the box. Following this, for neutralization in systems involving AChE seven Na+ ions (NaCl; 0.15 M) were added, while four and five Cl- ions were added in systems with BChE and BACE-1, respectively, the same was repeated for the complexes.

The APO forms were minimized by the Steepest Descent algorithm (SD) with 10,000 cycles, followed by the Conjugate Gradient algorithm (CG) with 1000 cycles. After the minimization steps, the complexes were then subjected to MD simulations. The equilibrium was performed during 1 ns, the heating step from 0 to 300 K during 1 ns, and the production step during 200 ns, with constant temperature (300 K) and pressure (1 bar).

The MD data were obtained in the isothermal–isobaric ensemble-NPT (number of particles, temperature and constant pressure) using periodic boundary conditions. Electrostatic and hydrophobic interactions were described by the PME method (Particle Mesh Ewald) [43] with a cut-off radius of 0.9 nm. The stability of the system was evaluated by analyzing the variation of the Root Mean Square Deviation (RMSD) value, Root Mean Square Fluctuation (RMSF) and turning radius with the aid of the RMS modules, RMSF and GYRATE, respectively, available in the GROMACS 5.1.2 package.

The number and permanence of hydrogen bonds (H) between the ligand and the residues of the binding site were calculated using the HBOND module available in GROMACS 5.1.2 and the HbMap2Grace program developed by Gomes and collaborators (unpublished data). As criteria for the identification of these bonds, the following parameters were evaluated: distance between donor and acceptor ≤ 3.5 Å and angle (α) between donor/acceptor/atom H ≤ 60${}^{\circ }$. The hydrogen bonds that remained for less than 10% of the simulation time were disregarded.

The representative structure of the ligand-macromolecule complex during the MD production phase was obtained from the grouping of similar conformations with the aid of the G_CLUSTER module available in the GROMACS 5.1.2 package, using the GROMOS union method [44], with cut-off points defined from the values of RMSD 1.0; 1.1; 1.2; 1.3; 1.4; 1.5 Å. The average structure of the most populous group was then selected to analyze the main interactions between the prioritized ligand and the targets.

## 3. Discussion and Results

### 3.1. Docking-Based Virtual Screening

Molecular docking is a receptor-based technique capable of increasing the success rate of virtual screening by evaluating the different binding modes of a molecule at the binding site, predicting the binding affinity and, therefore, indicating the compounds filtered in the Ligand-based Virtual Screening that are more likely to interact with targets [45]. In this context, a new methodology, called hierarchical or successive screening, has been proposed to systematically and progressively take advantage of the knowledge about ligands and their relationship with the structure of biological targets against more than one target [46,47]. Through this approach, 2119 fragments were successively evaluated, with 1091 having an affinity energy value of AChE < −7.1 Kcal/mol; 582 with an affinity energy value of BChE < 7.8 Kcal/mol; and 51 fragments having favorable scores for interactions with the active site of BACE-1 with a score bigger than 31.561 with respect to the median.

The LE of the selected fragments was calculated. This method compares molecules according to their average binding energy per atom. This concept provides an estimation of physicochemical properties that can be optimized to increase affinity for a target. The binding efficiency metric is recommended at all stages of drug discovery, starting with selecting a fragment or a screening success [31]. The binding efficiency is the ratio between the affinity energy value and the number of heavy atoms, with an optimal threshold of LE < −0.4 Kcal/mol. As a result, eleven, twelve, and fifteen fragments were selected for AChE, BChE, and BACE-1, respectively.

The intermolecular interaction between fragments and crystallographic ligand against molecular targets were evaluated using PyMOL™1.3 software [48]. Thus, as a result, four, seven, and eight fragments were selected for AChE, BChE, and BACE-1, respectively. In Figure 1, a comparison between the crystallographic ligand and a fragment for each target is illustrated. As can be observed, the pose of the fragment ZINC183800 selected for AChE (Figure 1A.1–2) shows the presence of residues Phe338, and Tyr337 in the catalytic site and also the residues Tyr72, Trp286 and Tyr341 in the peripheral anionic site, which participate in the adhesion of the substrate to the enzyme, thus ensuring the efficiency of the catalytic process [49,50]. In addition, the fragment ZINC183800 was selected for BChE (Figure 1B.1–2). This fragment shows the interaction with residues constituting the catalytic triad (Ser198, His438, and Glu325) [51]. Finally, the fragment ZINC36615 interacts with Asp32 and Asp228, the main residues involved in the catalytic process [52,53] of BACE-1 (Figure 1C.1–2).

### 3.2. Molecular Hybridization

The best fragments selected in previous stages against the selected molecular targets were joined through linkers using Marvin® Sketch 15.4.20 [30] and converted to 3D format on the SYBYL©-X 2.0 platform [22], resulting in a library containing thirteen molecular hybrids (Figure 2). It is worth noting that each union of fragments was selected with respect to affinity towards the targets, so the fragments that were in common were selected and joined to what acted on only one target. Thus, all of the hybrids have an affinity toward the three targets. The choice of linkers was based on the alignment of the hybrid to the triple pharmacophoric model.

The fragments used for the formation of the hybrids were the codes ZINC173351, ZINC183800, ZINC21985310, ZINC57379, ZINC16696635, ZINC9766, ZINC1261, ZINC13458, ZINC13597823, ZINC1464, ZINC13586781, ZINC36615, and ZINC4199922.

### 3.3. Physicochemical and Toxicological Filters

The use of receptor-based computational techniques plays an important role in identifying molecules capable of binding to three targets simultaneously. Therefore, the hybrids prioritized in the molecular docking step were further evaluated through Lipinski’s and Veber’s rules to evaluate their oral bioavailability and for potential mutagenicity through the Ames test that was used to discard molecules with toxic potential [34,35,54]. The results are summarized in Table 1. As can be observed, molecular hybrids positive for the AMES test were discarded; however, the others were considered for further steps, and it is known that about 6% of approved drugs do not obey the rules and are still bioavailable [55]. The remaining molecular hybrids (B, D, E, F, G, H, I, K and L) were then subjected to virtual screening by pharmacophoric model followed by molecular docking to confirm the presence of the necessary stereo-electronic requirements for the inhibition of three molecular targets.

### 3.4. Pharmacophore-Based Virtual Screening

Virtual screening based on pharmacophores can address the search of compounds with the same stereoelectronic characteristics as known active inhibitors/modulators. Thus, according to the previous report [16], the pharmacophoric models were generated using the GALAHADTM module, which seeks to identify the set of ligand conformations that have an optimal combination of pharmacophoric similarity, stereo overlap, and low-conformational energy [56]. From the training set, ten pharmacophoric models of triple inhibitors were generated. Pharmacophoric models generated by ligands with energetically unfavorable conformations were excluded [57,58]. In this context, four pharmacophoric models showed an energy penalty (Energy > 100.0 kcal/mol) and were excluded from the other validation steps. However, the exclusive use of this parameter was not enough to select a single pharmacophoric model.

The PARETO value, which represents a normalization of the values of the quality components of the pharmacophoric models, demonstrated that no pharmacophoric model was statistically superior to another [59], as all values were equal to zero. It was necessary to apply enrichment metrics. An important parameter to be evaluated in virtual screening methods is the ability to differentiate active compounds (true positives) from inactive compounds (false positives-decoys), based on the construction of the receiver operating characteristic curve (ROC curve) and the Area Under the ROC curve (AUC ROC) [60].

Additionally, another factor to be considered when evaluating the performance of pharmacophoric models is the predictive analysis of their ability to identify actives before identifying false positives and assigning higher scores to true positives in the early stages of the alignment process [25,61] with the help of early enrichment by Boltzmann-enhanced discrimination of ROC (BEDROC), given that only a small part of the molecule library will be acquired for biological tests.

Based on the AUC and BEDROC data, only one pharmacophoric model (AUC = 0.72 and BEDROC = 0.75) fulfils the necessary requirements for a reliable pharmacophoric model.

Molecular hybrids characterized by physical–chemical and toxicological filters were aligned with the best triple pharmacophoric model previously built and evaluated [16]. Molecular hybrid G was the only one that aligned to the triple pharmacophoric model and for this reason, it was selected (Figure 3).

The triple pharmacophoric model has a positively charged center, three hydrogen bond acceptor centers, and four hydrophobic centers, characteristics described as important for activity against the enzymes [25,62,63]. The presence of the positively charged atom is observed in compounds that can be metabolized by cholinesterases, being fundamental for the interaction with the aspartate residues present in the binding site (Asp74 for AChE and Asp70 for BChE). In addition, it can interact with the aspartyl dyad observed at the active site of BACE-1.

### 3.5. Molecular Docking

After selection of the hybrid better aligned with the triple pharmacophoric model, molecular docking was performed against the three pharmacological targets, using AutoDock Vina 1.1.2 [24], against AChE and BChE and GOLD 5.4.0 [28], against BACE-1 for the prediction of ligand affinity for the site and binding efficiency. After molecular docking, a visual inspection of the best-scoring pose and the crystallographic ligand was carried out for each target using PyMOL™1.3 [48] (Figure 4, Figure 5 and Figure 6).

Figure 4A shows the crystallographic ligand, dihydrotanshinone I (4M0E), which is stabilized at the binding site through a π-π bond with Trp286, the oxygen atom attached to the aromatic ring of the ligand acts as hydrogen bond acceptor with Phe295 and Tyr124. Hydrophobic interactions with residues Trp286 and Tyr341 are also observed. Upon evaluating the interactions of the molecular hybrid (Figure 4B), it was possible to observe a π-π bond between the ring attached to Tyr341, the ring of the molecular hybrid, and Trp286, whereas the hydrophobic interactions occur with Leu76, Tyr341, and Leu289. This profile is considered important for AChE inhibition since interactions of the same nature involving Trp286 are observed in AChE inhibitors with inhibitory activity at the nanomolar scale [64].

The crystallographic ligand of BChE (4BDS), tacrine, which was one of the first drugs to be used for the treatment of patients with AD, the heteroaromatic ring of the ligand establishes π-π stacking interactions with the aromatic nuclei of the residue Trp82 and also establishes hydrophobic interactions with residue Trp82. The intermolecular interaction map of the molecular hybrid (Figure 5B) shows that hydrophobic interactions occur with Asp70, Gly116, Trp82, Gln119, Thr120, Phe329, and Gly117. Hydrogen interactions also occur with Thr120, Glu197, Ser198. The interactions observed, both in the ligand and in the hybrid, with Trp82 residues present in the anionic site prevent the substrate from reaching the catalytic site. In the interaction map of the hybrid, it is also possible to observe an ionic interaction with Asp70, which is part of the peripheral site. In addition, other interactions with Thr120, Phe329 have been observed in other potent BChE inhibitors [65].

The crystallographic ligand of BACE-1 (6UWP) establishes hydrogen interactions between the nitrogen atoms of the thiazinemidine ring with the catalytic residues Asp32 and Asp228. A hydrogen interaction between the nitrogen of the side chain and Gly230 and the pyrimidine ring establishes a hydrophobic interaction with residue Val69. The molecular hybrid (Figure 6B) performed a π-π bond between the hybrid ring and the phenolic ring of Tyr71, and hydrophobic interactions occur with residues Arg128, Val69, Thr72, Gln73, Tyr71, Ile110, Trp115. Previous reports have shown that interactions with Asp32 and Asp228 are essential for biological activity. Interactions with Tyr71, which were observed in the molecular hybrid, are essential for inhibiting activity as they promote a change in conformation and prevent the substrate from reaching the catalytic site. Interactions with the Val69, Ile110, and Trp115 are also cited as important for inhibiting BACE-1 [66].

### 3.6. Molecular Dynamics

Molecular dynamics (MD) is a computational technique capable of approaching biological phenomena, using simple approximations based on Newtonian physics to simulate atomic movements, thus reducing computational complexity [18], and describing the variation of molecular behavior as a function of time [17]. The objective of MD in this study was to evaluate the stability of the molecular hybrid at the active sites of AChE, BChE, and BACE-1 and also to evaluate the pattern of interactions established in relation to molecular docking.

Initially, the systems used in the MD simulations were evaluated for structural stability through the root-mean-square deviation (RMSD), which provides information on the variation of atomic positions compared to the starting structure. Low RMSD values suggest the moment the system reaches stabilization [67].

The APO and complex systems (molecular hybrid linked to AChE, BChE and BACE-1) were evaluated for RMSD for the main chain (backbone) along the trajectory (Figure 7). As can be seen in Figure 7A, APO forms stabilized from 25 ns onwards, reaching few structural variations, and, therefore, the production phase was defined for data acquisition. In addition, as can be seen in Figure 7B, the MD routines for the complexes showed stabilization from 150 ns for AChE, BChE, and BACE-1. APO and complex systems have divergent simulation times, as the equilibrium may vary according to inherent characteristics of the binding between the protein and ligand. Thus, RMSD lower than 3.0 Å is considered appropriate [17].

RMSD analysis is considered a useful evaluation metric that considers protein flexibility. In addition, the behavior of the binding-site residues was observed to evaluate the atomic fluctuations of the residues individually through the root-mean-square fluctuation (RMSF) calculation. For this reason, residual fluctuation graphs were generated for the APO forms and compared with those of the respective complexes during the production phase of previously defined by MD (Figure 8).

As can be observed in Figure 8A, the atomic fluctuation of the complex with AChE (RMSF = 0.9 ± 0.4 Å) is statistically similar to the fluctuation of the APO form (RMSF = 1.2 ± 0.7 Å) (Figure 8A). The regions of the orthosteric site followed a similar fluctuation profile both in the complexed form and in the APO form, which demonstrates that there were no significant changes in the conformation of the site in the simulations. Region A corresponds to residues Tyr72, Asn74, and Trp86. Region B corresponds to residue Tye124. The regions identified by the letters C, H, and I had higher fluctuation. The regions C (region of residues Ala160-Glu166) and H (Asp494-Ala497) correspond to loop regions and region I (Leu540-Ala542) corresponds to the tip chain. All of them were characterized by conformational flexibility.

The BChE showed RMSF values of 1.0 ± 0.4 Å and 0.8 ± 0.4 Å for the APO form and of the complex, respectively, suggesting that the complex presents higher stability (Figure 8B). As BChE changes its conformation through coordinated movements to allow access to the substrate at the active site, the fluctuations are more evident for the APO to form at the active site regions (A, B, D, E). In the complex form, substrate entry is blocked, as movements are inhibited, and smaller fluctuations are observed. Regarding the C and F regions where the fluctuation peak is observed, in the C region (Ser210-Leu216) corresponds to a loop region, and in the F region (Pro527-Val529), corresponds to the end of the chain, which is characterized by conformational flexibility.

The BACE-1 exhibited RMSF values of 0.8 ± 0.6 Å and 0.9 ± 0.6 Å for the APO form and the complex, respectively, showing similarity in their values. The regions B and C that comprise the catalytic residues had low fluctuation, similar in the two systems, demonstrating that there were no significant changes in the conformation of the site in the simulations. Moreover, the higher fluctuations observed in the A (Pro-Val) and D regions (Val312-Ser315) can be attributed to the loop regions.

To understand the stabilization process of the systems, it is necessary to investigate the intermolecular interactions formed between the molecules and the amino acids at the binding site. Non-covalent interactions are essential for the maintenance of protein structure, and hydrogen bonds contribute to the maintenance of tertiary structure and protein folding as well as molecular recognition [68]. As this type of bond can be transient and directed, it is important that not only their occurrence be evaluated, but also their duration/permanence in the simulation time and by the pairs of atoms that make those interactions [46]. Thus, our findings showed a set of hydrogen bonds during the productive phase of the simulations in the active sites of AChE, BChE, and BACE-1 and the hybrid (Figure 9).

As can be observed in Figure 9, the AChE complex presents two hydrogen interactions between the N3 nitrogen of the hybrid and the ASN87 residue alternating as a donor (1.60%) and as an acceptor (22.16%); however, for a period greater than 10%, established. Residue interactions do not influence catalytic activity, but stability and folding, as can be seen with ASN289. It is known that interactions with these residues also occur with the cholinesterase inhibitor Donepezil [69].

In the BChE complex, interactions with permanence are higher than 10% between the N3 nitrogen of the hybrid and GLN71 (11.98%), and also between the N7 nitrogen of the hybrid and ASN289 (15.17%). Interactions with GLN71 are also present in the interaction map of the most representative structure in our study. Similar to ASN289 in AChE, GLN71 in BChE influences stability and folding as well.

In the BACE-1 complex, there were a total of six interactions of H, namely, between the N5 nitrogen of the hybrid and the GLN73 residues, in one the residue appears as a donor (1.00%) and in the other as an acceptor. (24.75%). There was an interaction between the N3 nitrogen of the hybrid and the residues of ASP32, in one the residue appears as a donor (8.98%) and as an acceptor (7.78%). There was also an interaction between the N3 nitrogen of the hybrid and the residues ASP228, in one the residue is as a donor (8.98%) and in another as an acceptor (88.42%). However, the hydrogen bonds between the residues and the hybrid are established for a period greater than 10%. Residues ASP32 and ASP228 are located in the active site of BACE-1 and are directly involved with the catalytic mechanism [35,41,53]. Hydrogen interactions with GLN73 have been observed in complexes with inhibitors of known activity against BACE-1, where it is possible to observe the enzyme in a closed conformation that denies access to the substrate, this interaction is also present in molecular docking.

Analysis of hydrogen bonding does not cover all the interactions that the hybrid can make in the active site of each target. Other interactions can happen and collaborate with the stability of the system. In order to evaluate these interactions, it was necessary to select a representative structure, where an analysis was carried out through an appropriate cut-off point that represented the conformational diversity of the complex during the simulation [70].

The analysis using different cut-off points pointed out that the most suitable RMSD value to be used was 1.1 Å since it presented a greater number of clusters with significant diversity and more than 50% of the conformations were presented in the three first groupings for the three systems [71].

The population of the three largest clusters (for the three targets) represented more than 50% of the total conformations of the systems during the production phase. Thus, it was possible to obtain the coordinates of the central structure for AChE (time = 162,700 ns) (Figure 10A), BChE (time = 178,400 s) (Figure 10B), and BACE-1 (time = 160,000 ns) (Figure 10C), which is representative of the most populous grouping. Furthermore, these clusters were the most prevalent during the trajectory of the productive phase of the MD simulation for the three targets.

The interaction map of the representative structure of the AChE complex shows hydrophobic interactions between the molecular hybrid and amino acid residues Val73, Tyr77, Trp286, Gln291 and Leu76 and also hydrogen bonding with Asn87 (Figure 10A). The interaction of hydrogen with Asn87 was also seen in the interaction established with N3 of the crystallographic ligand. Moreover, the intermolecular interaction map of the crystallographic ligand (Figure 4A) shows hydrophobic interaction with Trp286, and this is characteristic of AChE inhibitors phthalazine derivatives with biological activity [72].

The interaction map of the most representative structure for BChE, hybrid “G”, shows hydrophobic interactions occur with residues Gly117, Phe329, Val288, Trp82, Tyr332, Gln71, Thr120, Ser79, Trp460, Ser72 (Figure 10B). Moreover, π-stacking interactions occurred between rings [72], of the residues Tyr332 and Phe329. In addition, a previous report showed that the Val288 and Tyr332 residues are located in the region close to the catalytic residues [73,74]), which are responsible for interacting with the substrate at the active site. Furthermore, the interaction with Trp82 was also observed in the crystallographic ligand (Figure 5A) and prevents the substrate from reaching the catalytic site [52,72].

The interaction map of BACE-1 with hybrid “G” shows hydrogen interactions established with the catalytic residue Aps228 (Figure 10C), which is also present in the crystallographic ligand interaction map (Figure 6A). This interaction was also repeated in Hbond with a residence time longer than 80%. Interactions with Asp32 and Asp228 residues are essential for biological activity, as they are present at the catalytic site. π-stacking interactions occur with residues Tyr198 and Phe108. Hydrophobic interactions with the rings of this group occur with residues Tyr71, Thr72, Ile226, Val332, Ile110, and Phe108. Interactions with the amino acid residue Tyr71 are essential for inhibiting activity as they promote a change in the conformation of the site and prevent the substrate from reaching the site, residues Phe108 and Ile226 contribute to the stability of the ligand-macromolecule complex and they have been observed in previously described MD simulations [75,76]. Interactions with the amino acid residue Ile110A are cited as important for the inhibition of BACE-1 [66].

According to the data obtained, molecular hybrid G has the essential stereo-electronic requirements for affinity with the three targets, AChE, BChE, and BACE-1.

Due to hepatotoxicity being one of the main side effects associated with anti-AD drugs, the cytochrome-inhibition values related to Molecular hybrid G were then calculated. It used the same pkCSM server [33] that characterized and evaluated the hybrids in the Physicochemical Filters step. The data obtained were that the compound acts as a CYP2D6 substrate; CYP3A4 substrate; CYP2C19 inhibitor; CYP2C9 inhibitor and CYP3A4 inhibitor. From the analysis of the results, it is necessary to emphasize the attention warranted toward the concomitant use of other drugs, as there may be drug interactions.

## 4. Main Limitations of This Work

The use of computational techniques to select molecules obeying proper stereo-electronic requirements and binding at the target site is important for the identification of inhibitors. Despite the advantages offered by computational techniques, some limitations are inherent.

Despite compound G, in addition to exhibiting some physical–chemical limitations and high-structural complexity, it presents important molecular interactions against the inhibition of targets.

However, analyses carried out by [77], on a set of compounds in addition to the chemical space proposed by the rules of Lipinski and Veber, demonstrated that they were orally bioavailable. The observation that ligands for difficult targets have higher MW and greater lipophilicity than highly explored classes lends further support. Synthetic chemistry and purification challenges associated with iterative and exhaustive modifications of complex molecules must also be considered, especially for natural product leads. Recent and future advances in synthetic methods may reduce this concern [78].

## 5. Conclusions

This study designed a hybrid compound capable to inhibit the three molecular targets proposed to be associated with the disease. This goal was reached using a set of molecular modeling tools, docking, virtual screening, and molecular dynamics simulations. Thus, the use of a generic algorithm by GOLD, which has the ability to generate satisfactory solutions and score functions, allowed molecular docking against BACE-1. In addition, AutoDock Vina allowed docking to AChE and BChE. In other words, our study showed that docking methodologies are structure dependent; therefore, requiring evaluations such as redocking and ROC curve.

Virtual screening followed by the analysis of binding efficiency selected eleven fragments for AChE, twelve fragments for BChE, and fifteen fragments for BACE-1. Further analysis of the position of these fragments in the active site reduced those numbers to four fragments for AChE, seven fragments for BChE, and eight fragments for BACE-1.

The interactions of these fragments occurred with the same residues of the active sites of each enzyme observed with the crystallographic ligands. Molecular hybridization of those fragments generated a structure capable of an alignment with the triple pharmacophoric model with QFIT > 27.00. During MD simulations, it was shown that the presence of the hybrid G at the active site in all complexes (for the three targets), stabilized the system, resulting in lower residue fluctuations and lower RMSD values in the production phases of the complexes when compared to the APO forms.

The computational techniques described, when employed in a hierarchical process, enabled the selection of molecules with proper stereo-electronic requirements for an affinity towards the three targets, and suggest that molecular hybrid “G” bears the potential to be a triple inhibitor of those enzymes. Future steps involve the synthesis and evaluation of biological activity for validation of our triple inhibitor model. Finally, our findings suggest a multi-target molecule for the treatment for Alzheimer’s disease, which can ultimately lead to an increased quality of life for affected patients.

## Figures and Tables

**Figure 1 pharmaceuticals-16-00880-f001:**
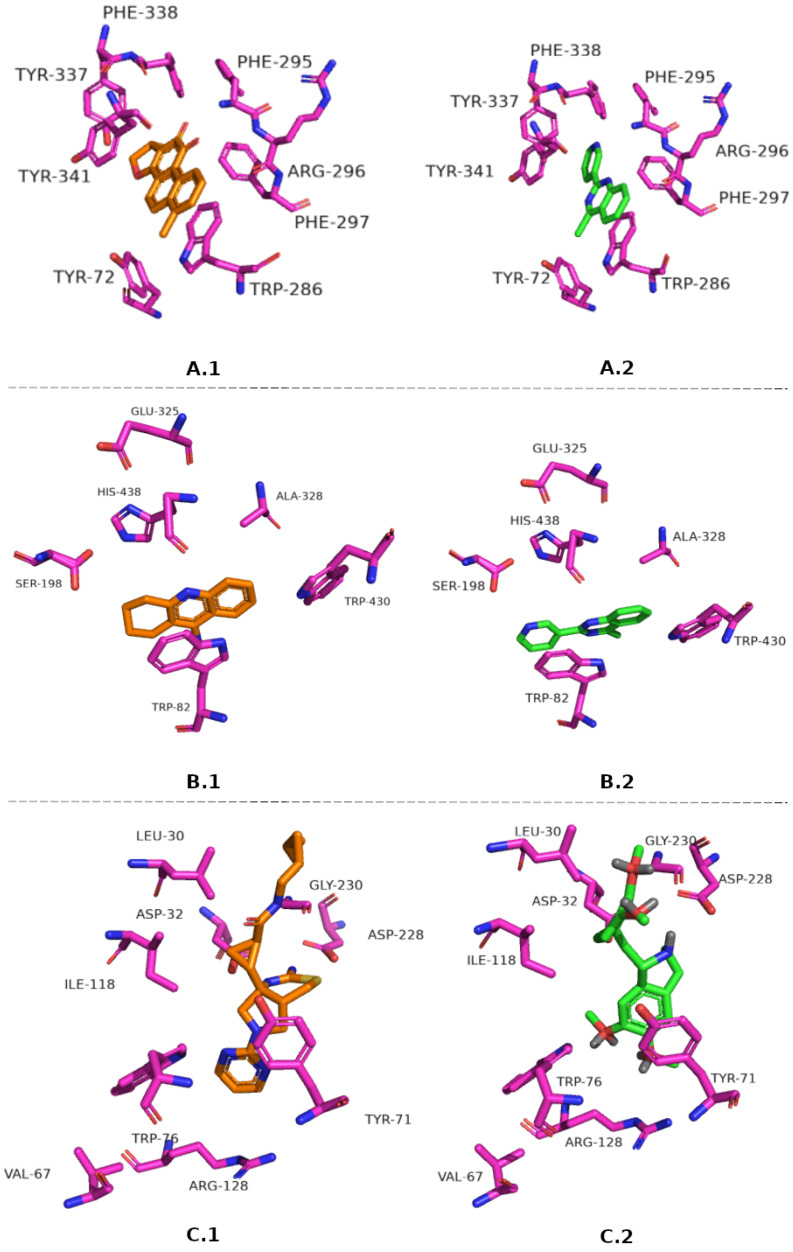
PosePosition of crystallographic ligands and fragments in the active targets’ site, AChE Dihydrotanshinone I and ZINC183800 (**A.1**–**2**); BChE Tacrine and ZINC183800 (**B.1**–**2**), and BACE-1 QKA and ZINC36615 (**C.1**–**2**).

**Figure 2 pharmaceuticals-16-00880-f002:**
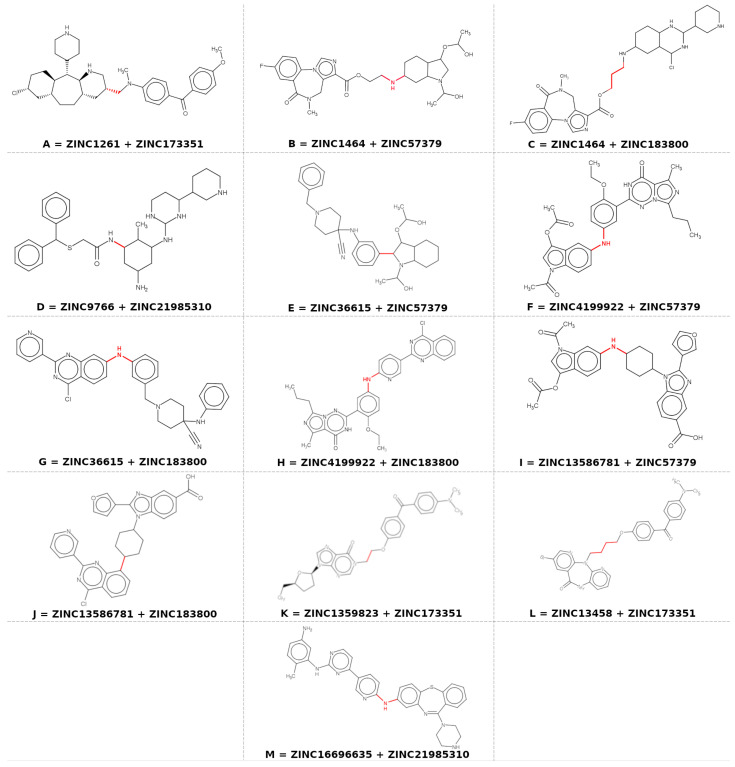
Fragments and chemical structure of the hybrids (Red bonds are used to highlight the linker).

**Figure 3 pharmaceuticals-16-00880-f003:**
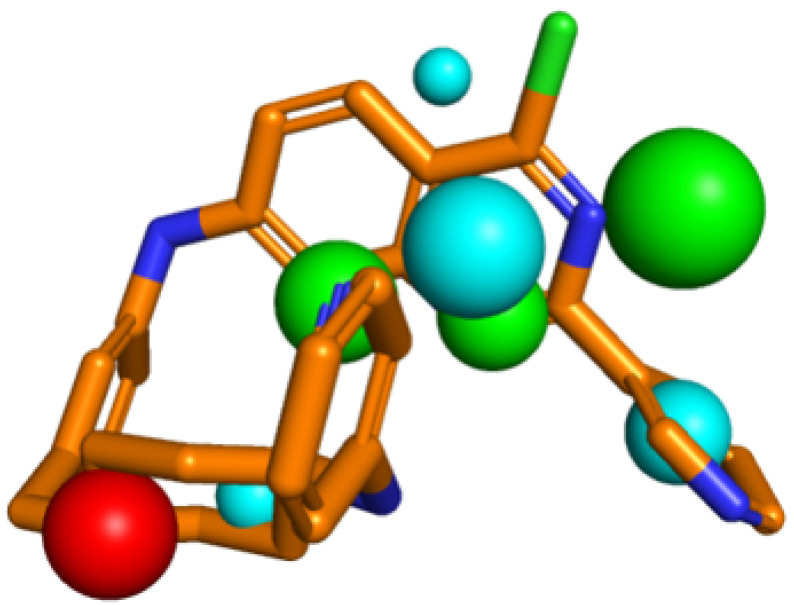
Representation of the triple model and molecular hybrid overlap (cyan spheres: hydrophobic centers; green: H-bond acceptor points; and red: positively charged center).

**Figure 4 pharmaceuticals-16-00880-f004:**
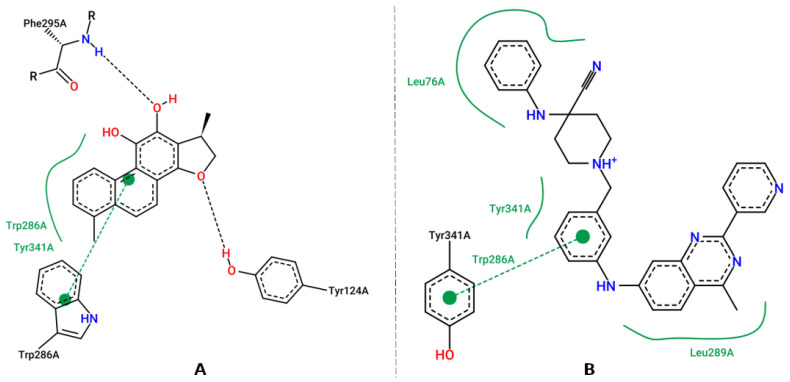
Representation of the intermolecular interactions of the crystallographic ligand dihydrotanshinone I (**A**), and the best-scored pose for AChE with molecular hybrid (**B**).

**Figure 5 pharmaceuticals-16-00880-f005:**
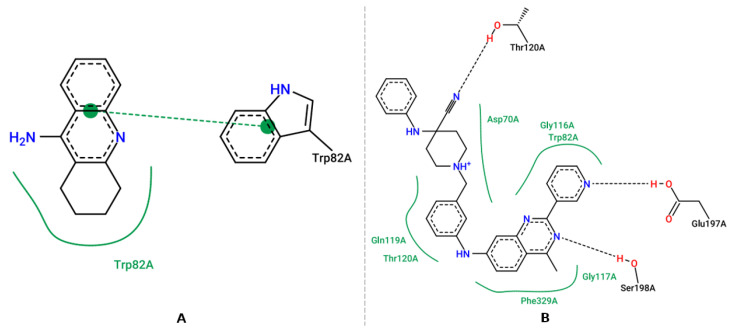
Representation of the intermolecular interactions of the crystallographic ligand tacrine (**A**), and the best-scored pose for BChE with molecular hybrid (**B**).

**Figure 6 pharmaceuticals-16-00880-f006:**
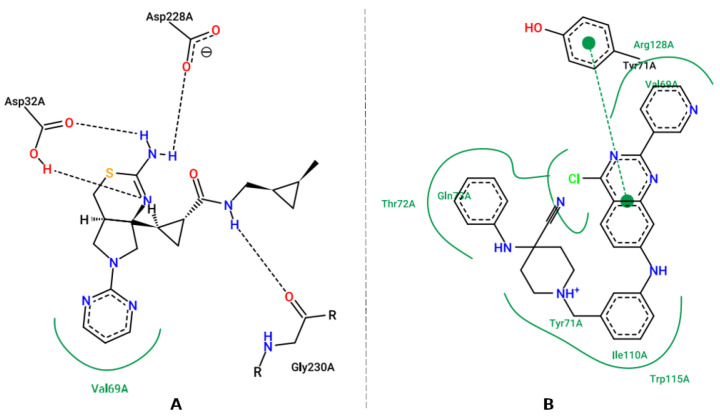
Representation of the intermolecular interactions of the ligand (1R,2R)-2-[(4aR,7aR)-2-amino-6-(pyrimidin-2-yl)-4a,5,6,7-tetrahydropyrrolo[3,4-d][1,3]thiazin-7a(4H)-yl]-N-[(1R,2R)-2-methylcyclopropyl]methylcyclopropane-1-carboxamide (**A**), and the highest scoring pose for BACE-1 with molecular hybrid (**B**).

**Figure 7 pharmaceuticals-16-00880-f007:**
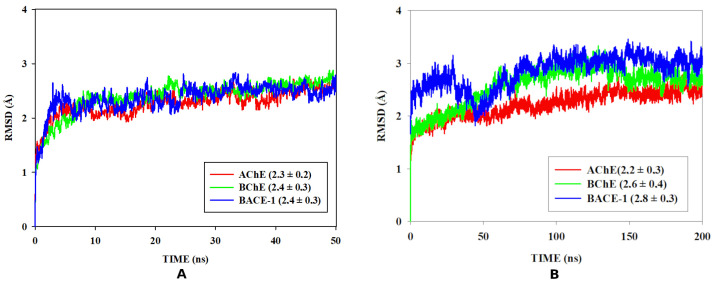
RMSD (backbone) of APO structures (**A**) and complexes of AChE, BChE and BACE-1 with molecular hybrid (**B**) during molecular dynamics.

**Figure 8 pharmaceuticals-16-00880-f008:**
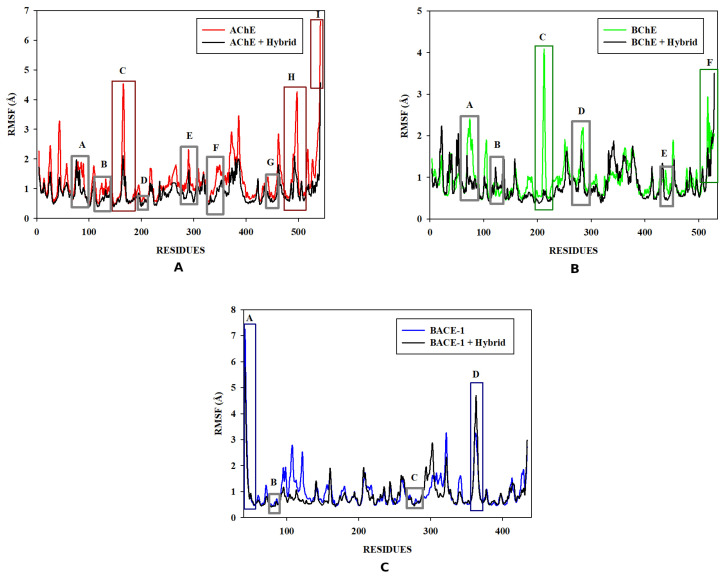
RMSF (Å) (backbone) of the APO structure and AChE complex with molecular hybrid during the productive phase of molecular dynamics (**A**). RMSF (Å) (backbone) of the APO structure and BChE complex with molecular hybrid during the productive phase of molecular dynamics (**B**). RMSF (Å) (backbone) of the APO structure and BACE-1 complex with molecular hybrid during the productive phase of molecular dynamics (**C**). Gray highlights correspond to active site regions.

**Figure 9 pharmaceuticals-16-00880-f009:**
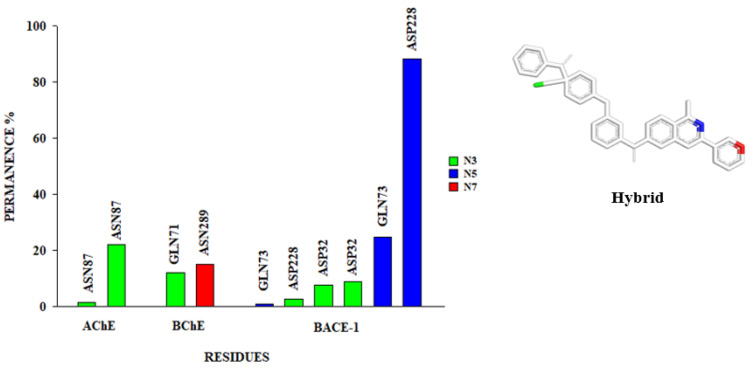
Time of permanence of hydrogen interactions (Hbond) in the active site of AChE, BChE and BACE-1 during the productive phase and identification of the involved pairs. In the chart legend; N: nitrogen atoms of the hybrid.

**Figure 10 pharmaceuticals-16-00880-f010:**
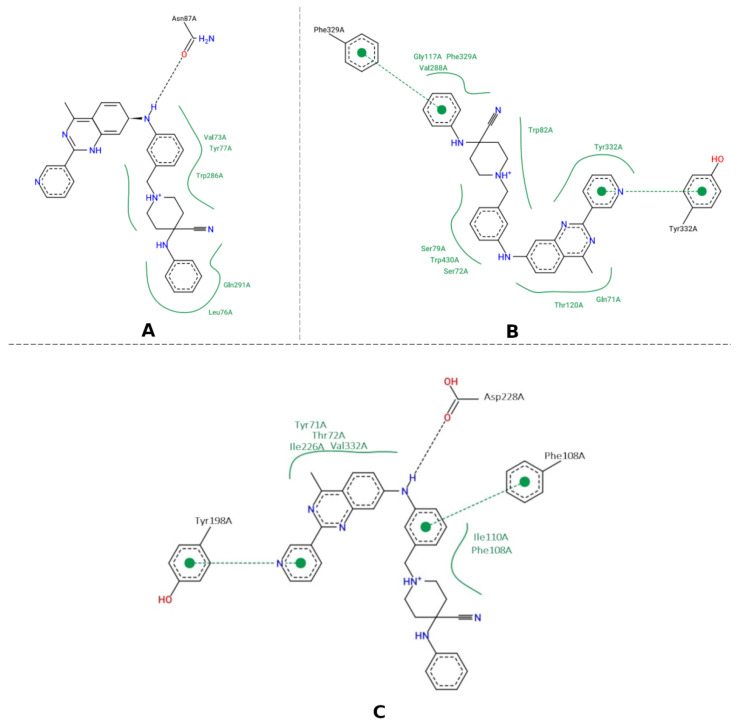
Representation of the interactions of the hybrid complex plus the AChE target obtained from the representative structure during the productive phase (**A**). Representation of the interactions of the hybrid complex with BChE obtained from the representative structure during the productive phase (**B**). Representation of the interactions of hybrid “G” with BACE-1 obtained from the representative structure during the productive phase (**C**).

**Table 1 pharmaceuticals-16-00880-t001:** Physical–chemical and toxicological filters.

Molecule	MW (g/mol)	HBD	HBA	cLog P	PSA (Å${}^{2}$)	RB	HBD + HBA	AMES
A	578.24	2	5	6.39	251	7	7	YES
B	545.61	3	10	1	226	8	13	NO
C	588.13	4	9	2.35	244	7	12	YES
D	550.08	6	7	2.94	238	9	13	NO
E	518.70	3	7	6.63	227	8	10	NO
F	542.60	2	10	5.03	229	8	12	NO
G	546.08	2	7	7.06	238	7	9	NO
H	567.05	2	9	6.15	240	8	11	NO
I	540.58	2	9	6.13	229	6	11	NO
J	1098.0	1	15	14.94	466	11	16	NO
K	503.56	1	10	2.64	213	9	11	NO
L	586.49	1	7	6.10	232	9	8	NO
M	585.74	4	10	6.37	254	5	14	YES

MW = molecular weight; HBD = hydrogen bond donor; HBA = hydrogen bond acceptor; cLog P = octanol–water partition coefficient; PSA = polar surface area; RB = rotatable bonds. AMES: YES = potential carcinogenicity, NO = no concerns about carcinogenicity.

## Data Availability

Data is contained within the article.

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
