# Peer review of "Molecular Multi-Target Approach for Human Acetylcholinesterase, Butyrylcholinesterase and β-Secretase 1: Next Generation for Alzheimer’s Disease Treatment"

_pharmaceuticals, 2023, doi:10.3390/ph16060880_

Round 1
Reviewer 1 Report
The paper is well written. It concerns AD, specifically screening against AChE, BChE and BACE-1, was done. Included are 9 figures and all the paper's relevant sections. Conclusions: This study designed an inhibitor for three molecular targets at the same biological pathway of AD. This goal was reached using a set of molecular modeling tools, docking, virtual screening, and molecular dynamics simulations. Thus, the use of a generic algorithm.
by GOLD, which has the ability to generate satisfactory solutions and score functions
allowed molecular docking against BACE-1. Statistics were also applied. They also cite Prof. John Hardy.
the main question addressed by the research-- identification of multi-target drugs to treat AD
the subject area compared with other published material
molecular coupling against the 3D structures 73
of the targets, AChE, BChE and BACE-1, respectively. 74
The 3D structures of the targets, AChE, BChE, and BACE-1, were obtained from the 75
Protein Data Bank (PDB)
The MD data were obtained in the isothermal-isobaric ensemble - NPT (number of 162
particles, temperature and constant pressure) using periodic boundary conditions. Elec- 163
trostatic and hydrophobic interactions were described by the PME method
The representative structure of the ligand-macromolecule complex during the MD 176
production phase was obtained from the grouping of similar conformations
Author Response
Dear Reviewer,
The comments and criticisms raised by referees will contribute to improving the overall scientific impact of the manuscript. The article was fully revised to improve all aspects. All in all, we agree with all comments, and we have already made all the requested changes as can be seen in the revised manuscript. In general, the authors acknowledge the comments.
Sincerely,
Reviewer 2 Report
Dear authors,
I consider his work entitled “Molecular Multi-target Approach for Human Acetylcholinesterase, Butyrylcholinesterase, and β-secretase 1: Next Generation for Alzheimer’s Disease Treatment" interesting and transcendent. However, major and minor changes are necessary to incorporate.
I share some comments about it that I hope will be useful:
** Major comments:
1. Methods section – 2.2: Please, provide the list of fragments used to create the hybrid pharmacophore model. And, how many fragments were used? Are these fragments obtained from a database?
2. Methods section – 2.5: Although the docking protocols have been previously described and validated, I consider it necessary to add a detailed description of the molecular docking methodologies used in this work.
3. Methods section – 2.6: What is the main reason to select this MD server and this forcefield? Respectfully, I consider that exist a lot of other (and better) conditions for doing MD calculation. For example, Desmond, GROMACS, NAMD, OpenMM, AMBER, etc., generate higher-quality results. Please, justify why select this server, and add references that support the use of this server for these kinds of targets.
4. Methods section – 2.6: Again, please justify the use of the SPC water model to describe the MD of these kinds of targets. Why not TIP3, which is a low computational cost water model, but with better "molecular resolution"?
5. Results and Discussion – 3.2: Not is clear how to select the linkers used to join the fragments. Please, clarify this methodological point.
6. Results and Discussion – 3.3: Why select this PSA criterion? I understand that the drugs to pretend to act in the central nervous system must cross the encephalic membrane. Whit this in mind, and consider that the molecules with values > 90 Angstroms do not have the physicochemical features to cross this membrane... Respectfully, I do not understand why define 250 Angs to benchmark. Please, justify, remove, or adapt this criterion.
7. Methods, Results, and Discussion sections: As you mention in the introduction section, hepatotoxicity is one of the main side effects associated with anti-AD drugs. I recommend calculating/predicting the possible hepatotoxic issues related to the proposed drugs in this work, for example... the prediction of their cytochromes inhibition values.
8. Results and Discussion sections: Discussion section: I strongly recommend adding a brief section about the "main limitations of this work", which talks about 1) the hypothetical physicochemical limitations of the "G" compound to cross hematoencephalic membrane; 2) The high structural complexity of G compound; 3) their possible off-targets bioactivity, and; 4) and about the general issues to resolve around polypharmacological drug design to central nervous system targets. I share with you some references, that could be useful:
https://doi.org/10.3389/fnins.2016.00265
http://dx.doi.org/10.2174/0929867326666191212103330
https://doi.org/10.12688/f1000research.124160.1
** Minor comments:
1. Methods section – Line 73: Change the expression "molecular coupling" to "molecular docking".
2. Methods section – 2.4: Please, describe in more detail the QFIT metric, and add a reference that supports this.
3. Methods section – 2.6: What concentration of Na2+ and Cl- was added to neutralize the MD system?
4. Results and Discussion section – Line 194: Please add the metric units of the value "31.561".
5. Results and Discussion section – Line 231: Change the expression "ASP" to "PSA".
6. Conclusion section: I strongly suggest removing this last sentence "Finally, our findings suggest a new treatment for Alzheimer’s disease, which can increase the patient’s quality of life." Respectfully, I consider that these results only could contribute to the identification of new poly-drugs against AD, but not directly to the creation of a new AD treatment.
Author Response
Dear Reviewer,
The comments and criticisms raised by referees will contribute to improving the overall scientific impact of the manuscript. The article was fully revised to improve all aspects. All in all, we agree with all comments, and we have already made all the requested changes, as can be seen in the revised manuscript. In general, the authors acknowledge the comments. In relation to reviewer’s comments, several points need to be highlighted as follows:
Comment (1) “Please, provide the list of fragments used to create the hybrid pharmacophore model. And, how many fragments were used? Are these fragments obtained from a database?”
Answer: These data were adequately inserted in the manuscript.
Comment (2) “Although the docking protocols have been previously described and validated, I consider it necessary to add a detailed description of the molecular docking methodologies used in this work.”
Answer: Docking protocols used in this manuscript were previously published at Mascarenhas et al., 2020 and Bomfim et al., 2021. However, data from docking were inserted in the manuscript as can be seen in the text steps as 2.1.
Comment (3) “What is the main reason to select this MD server and this forcefield? Respectfully, I consider that exist a lot of other (and better) conditions for doing MD calculation. For example, Desmond, GROMACS, NAMD, OpenMM, AMBER, etc., generate higher-quality results. Please, justify why select this server, and add references that support the use of this server for these kinds of targets.”
Answer: I did not understand this comment because GROMACS server was applied to MD routine. Even though there are several good platforms for MD, the area is not mature enough to determine which is the best. GROMACS is a widely used molecular dynamics simulation package, and the GROMACS96 54A7 force field is implemented within this software. This ensures compatibility and easy integration with other GROMACS features and tools. GROMACS96 54A7 force field has been available for a long time and has been widely used by the scientific community. The force field has been extensively tested against experimental data, such as NMR spectroscopy, X-ray crystallography, and thermodynamic measurements. It has been shown to reproduce key experimental observables, including protein structures, folding free energies, and protein-ligand interactions. In addition, GROMACS is designed to efficiently utilize modern high-performance computing (HPC) architectures, including parallel processing capabilities, which allow simulations to be performed on a large number of processors, as we used. Furthermore, it incorporates state-of-the-art algorithms for treating long-range electrostatic interactions, such as the Particle Mesh Ewald (PME) method, and it provides options for including various types of molecular interactions, such as bonded and non-bonded interactions, implicit solvation models, and advanced sampling techniques. Moreover, GROMACS is an open-source software, which means that the source code is freely available and can be modified and extended by the user or the scientific community, Topology Builder (gmx2pdb) and GROMACS Analysis Tools (gmx analyze). These tools simplify the process of preparing input files, running simulations, and analyzing the results, even for users who may not have extensive programming skills. Finally, GROMACS has a large and active user community, including researchers from various disciplines, who contribute to its development and provide support. This active community ensures that GROMACS remains up-to-date, well-maintained, and responsive to the evolving needs of the scientific community. The references section contain some Gromacs server with Alzheimer targets studies.
Comment (4) “Again, please justify the use of the SPC water model to describe the MD of these kinds of targets. Why not TIP3, which is a low computational cost water model, but with better "molecular resolution"?
Answer: The choice of water model for molecular dynamics (MD) simulations, especially when studying proteins, depends on various factors, including the research objectives, system size, and available computational resources. While TIP3 is a commonly used water model, the SPC (Simple Point Charge) water model is also widely employed in MD simulations involving proteins. Here are some justifications for using the SPC water model in protein simulations:
First, The SPC water model has a long-standing history in MD simulations and has been extensively used in protein research. This extensive usage means that there are established protocols, force fields, and empirical data available for the SPC model, making it easier to compare and reproduce results across different studies; Second, reproduction of Experimental Observations: The SPC model has been shown to reproduce various experimental properties of water, such as the density, dielectric constant, and radial distribution functions. This makes it a reliable choice for studying water-protein interactions and their impact on protein structure and dynamics; Third, the balance of Accuracy and Computational Cost: The SPC model strikes a balance between accuracy and computational efficiency. While it is more computationally demanding than the TIP3 model, it provides a more detailed description of water molecules by considering their partial charges and explicitly accounting for the molecular structure. This increased accuracy can be crucial when studying the specific interactions between water molecules and protein residues; Fourth, Polarizable Force Fields: The SPC model is compatible with polarizable force fields, which can capture the dynamic polarization of water molecules and improve the accuracy of simulations. Polarizable force fields are particularly useful when studying the electrostatic interactions between proteins and water, as they can account for the polarization effects that occur due to the presence of charges in the protein; Finally, specific Protein-Water Interactions: The SPC model has been successfully employed in numerous studies focusing on protein hydration and water-mediated interactions. It has been shown to reproduce experimental observations related to the hydration structure around proteins and the energetics of protein-water interactions.
In summary, TIP3 This model uses a rigid geometry of water molecules. Differently, SPC model assumes an ideal tetrahedral shape (HOH angle of 109.47°) instead of the observed angle of 104.5° and adds an average polarization correction to the potential energy function, which improves the results.
Comment (5) “Not is clear how to select the linkers used to join the fragments. Please, clarify this methodological point.”
Answer: Inserted in the text, choice of linkers was based on the alignment of the hybrid to the triple pharmacophoric model.
Comment (6) “Why select this PSA criterion? I understand that the drugs to pretend to act in the central nervous system must cross the encephalic membrane. Whit this in mind, and consider that the molecules with values > 90 Angstroms do not have the physicochemical features to cross this membrane... Respectfully, I do not understand why define 250 Angs to benchmark. Please, justify, remove, or adapt this criterion.”
Answer: According to Weber's rule, molecules that have a PSA ≤ 140 Å should have the potential to be orally bioavailable, meaning the proposed cutoff by the reviewer is included in the cutoff proposed in this study. Additionally, the physicochemical filter was used only to characterize the hybrids.
Comment (7) “Methods, Results, and Discussion sections: As you mention in the introduction section, hepatotoxicity is one of the main side effects associated with anti-AD drugs. I recommend calculating/predicting the possible hepatotoxic issues related to the proposed drugs in this work, for example... the prediction of their cytochromes inhibition values.”
Answer: These data were calculated by pkCSM server and inserted in the manuscript.
Comment (8) “I strongly recommend adding a brief section about the "main limitations of this work", which talks about 1) the hypothetical physicochemical limitations of the "G" compound to cross hematoencephalic membrane; 2) The high structural complexity of G compound; 3) their possible off-targets bioactivity, and; 4) and about the general issues to resolve around polypharmacological drug design to central nervous system targets. I share with you some references, that could be useful:
https://doi.org/10.3389/fnins.2016.00265 http://dx.doi.org/10.2174/0929867326666191212103330 https://doi.org/10.12688/f1000research.124160.1”
Answer: The authors agree with the creation of this section. However, the limitations of computational techniques are already known and we understand that these aspects need to be made clearer in the text to avoid confusion regarding the scope of the computational data and its findings. For this reason, the references were useful to rewrite the discussion of study.
Comment (8) “I strongly suggest removing this last sentence "Finally, our findings suggest a new treatment for Alzheimer’s disease, which can increase the patient’s quality of life." Respectfully, I consider that these results only could contribute to the identification of new poly-drugs against AD, but not directly to the creation of a new AD treatment.”
Answer: The authors agree to propose and rewrite the paragraph in the manuscript. All minor comments were adequately inserted and highlighted in the manuscript.
Comment (4): Results and Discussion section – Line 194: Please add the metric units of the value "31.561".
Answer: GOLD is the validated, configurable protein–ligand docking software for expert drug discovery. For virtual screening through to lead optimization. The best ability to predict the best poses based on the best scores among the rest methods.
The functions obtained by GOLD are a normalization of the enrichment factor, that is, there is no unit of measurement. Since the score is based on the best scores obtained for each function, it uses a consensual model for the methods, in order to obtain an average score, which is used to evaluate the staging of the poses and thus result in the value of the function.
As described previously, we carried out all suggestions required by reference. All of them were welcome and contributed to improve the quality of manuscript. Hence, we discussed step by step and included all considerations in the text property highlighted. Thus, please, consider this paper for publication. We are sure that this version of the manuscript reached the exigence level required by the Pharmaceuticals audience.
REFERENCES
Mascarenhas, A.M.S.; de Almeida, R.B.M.; de Araujo Neto, M.F.; Mendes, G.O.; da Cruz, J.N.; Dos Santos, C.B.R.; Botura, M.B.; Leite, F.H.A. Pharmacophore-based virtual screening and molecular docking to identify promising dual inhibitors of human acetylcholinesterase and butyrylcholinesterase. Journal of Biomolecular Structure and Dynamics. 2021. doi: https://doi.org/10.1080/07391102.2020.1796791
do Bomfim, M.R.; Barbosa, D.B.; de Carvalho, P.B.; da Silva, A.M.; de Oliveira, T.A.; Taranto, A.G.; Leite, F.H.A. Identification of potential human beta-secretase 1 inhibitors by hierarchical virtual screening and molecular dynamics. Journal of Biomolecular Structure and Dynamics. 2022, doi: https://doi.org/10.1080/07391102.2022.2069155.
Gerben, S. R.; Lemkul, J. A.; Brown, A. M.; Bevan, D. R. Comparing atomistic molecular mechanics force fields for a difficult target: a case study on the Alzheimer’s amyloid β-peptide. Journal of Biomolecular Structure and Dynamics. 2013. doi: https://doi.org/10.1080/07391102.2013.838518
Gupta, M. K.; Vadde, R. In silico identification of natural product inhibitors forγ‐secretase activating protein, a therapeutic target for Alzheimerʼs disease. Journal of Cellular Biochemistry. 2018. doi: https://doi.org/10.1002/jcb.28316
Mishra, C. B.; Kumari, S.; Manral, A.; Prakash, A.; Saini, V.; Lynn, A. M.; Tiwari, M. Design, synthesis, in-silico and biological evaluation of novel donepezil derivatives as multi-target-directed ligands for the treatment of Alzheimer's disease. European Journal of Medicinal Chemistry. 2017. doi: https://doi.org/10.1016/j.ejmech.2016.09.057
Berendsen, H. J. C.; Grigera, J. R.; Straatsma, T. P. The missing term in effective pair potentials. J. Phys. Chem. 1987, doi: https://doi.org/10.1021/j100308a038
Bredenberg, J.; Mark, P.; Nilsson, L. CHAPTER 6 - Solvent effects on biomolecular dynamics simulations: A comparison between TIP3P, SPC and SPC/E water models acting on the Glucocorticoid receptor DNA-binding domain. Modern Methods for Theoretical Physical Chemistry of Biopolymers, 2006, doi: https://doi.org/10.1016/B978-044452220-7/50070-8
Jorgensed, W. L. Transferable Intermolecular Potential Functions for Water, Alcohols, and Ethers. Application to Liquid Water. J. Am. Chem. 1981, 103 doi: 10.1021/ja00392a016
Park, C.; Robinson, F.; Kim, D. On the Choice of Different Water Model in Molecular Dynamics Simulations of Nanopore Transport Phenomena. Membranes. 2022, doi: https://doi.org/10.3390/membranes12111109
Cole, J.; Nissink, J. W. M.; Taylor, R. Protein-Ligand Docking and Virtual Screening with GOLD. In B. Shoichet & J. Alvarez (Eds.), Virtual screening in drug discovery. Taylor & Francis CRC Press. 2005, doi: 10.1201/9781420028775.ch15
Round 2
Reviewer 2 Report
Dear authors,
I consider that all previous major and minor comments were resolved adequately. I consider that your manuscript is ready to be published.